# Classical Xanthinuria in Nine Israeli Families and Two Isolated Cases from Germany: Molecular, Biochemical and Population Genetics Aspects

**DOI:** 10.3390/biomedicines9070788

**Published:** 2021-07-07

**Authors:** Hava Peretz, Ayala Lagziel, Florian Bittner, Mustafa Kabha, Meirav Shtauber-Naamati, Vicki Zhuravel, Sali Usher, Steffen Rump, Silke Wollers, Bettina Bork, Hanna Mandel, Tzipora Falik-Zaccai, Limor Kalfon, Juergen Graessler, Avraham Zeharia, Nasser Heib, Hannah Shalev, Daniel Landau, David Levartovsky

**Affiliations:** 1Sourasky Medical Center, Clinical Biochemistry Laboratory, Tel Aviv 6423906, Israel; ayala.lagziel@gmail.com (A.L.); meirav.shtauber@gmail.com (M.S.-N.); zh.vicky@gmail.com (V.Z.); mariuss@bezeqint.net (S.U.); 2Department of Human Molecular Genetics and Biochemistry, Sackler School of Medicine, Tel Aviv University, Tel Aviv 69978, Israel; 3Department of Plant Biology, Braunschweig University of Technology, 38114 Braunschweig, Germany; florian.bittner@julius-kuehn.de (F.B.); rump@s-r-consulting.com (S.R.); s.wollers@tu-braunschweig.de (S.W.); bettina.bork@julius-kuehn.de (B.B.); 4Federal Research Center for Cultivated Plants, Julius Kuehn Institute, 06484 Quedlinburg, Germany; 5Department of History, Philosophy and Judaic Studies, The Open University of Israel, Raanana 43107, Israel; mustafa@openu.ac.il; 6SRConsulting, 31319 Sehnde, Germany; 7Department of Genetics and Metabolic Diseases, Ziv Medical Center, Safed 13100, Israel; h_mandel@rambam.health.gov.il; 8Rambam Medical Center, Metabolic Unit, Meyer Children’s Hospital, Haifa 3109601, Israel; 9The Ruth and Bruce Rappaport Faculty of Medicine, Technion-Israel Institute of Technology, Haifa 3109601, Israel; 10Institute of Human Genetics, Western Galilee Hospital-Nahariya, Nahariya 22100, Israel; falikmd.genetics@gmail.com (T.F.-Z.); LimorK@gmc.gov.il (L.K.); 11The Azrieli Faculty of Medicine, Bar-Ilan University, Ramat-Gan 5290002, Israel; 12Department of Pathological Biochemistry, Medicine III, Medical Faculty, Technische Universitaet Dresden, 01062 Dresden, Germany; Juergen.Graessler@uniklinikum-dresden.de; 13Day Hospitalization Center, Schneider Children’s Hospital, Petach-Tikva 4920235, Israel; azeharia@walla.co.il; 14Department of Pediatrics, Sackler School of Medicine, Tel Aviv University, Tel Aviv 69978, Israel; Danny_L@clalit.org.il; 15Medical Clinic, Clalit Health Services, Tuba-Zangariyye 1231000, Israel; nasserha@clalit.org.il; 16Department of Pediatrics, Soroka Medical Center, Beer Sheva 8457108, Israel; hannash@clalit.org.il; 17Faculty of Health Sciences, Ben Gurion University of the Negev, Beer Sheva 10455, Israel; 18Schneider Children’s Medical Center, Nephrology Institute, Petach-Tikva 4920235, Israel; 19Department of Rheumatology, Tel Aviv Sourasky Medical Center, Tel Aviv 6423906, Israel; dlevartovsky@gmail.com; 20Department of Rheumatology, Sackler School of Medicine, Tel Aviv University, Tel Aviv 69978, Israel

**Keywords:** xanthinuria, *XDH*, *MOCOS*, heterologous protein expression, Yemenite Jews, Arabs, founder effects

## Abstract

Classical xanthinuria is a rare autosomal recessive metabolic disorder caused by variants in the *XDH* (type I) or *MOCOS* (type II) genes. Thirteen Israeli kindred (five Jewish and eight Arab) and two isolated cases from Germany were studied between the years 1997 and 2013. Four and a branch of a fifth of these families were previously described. Here, we reported the demographic, clinical, molecular and biochemical characterizations of the remaining cases. Seven out of 20 affected individuals (35%) presented with xanthinuria-related symptoms of varied severity. Among the 10 distinct variants identified, six were novel: c.449G>T (p.(Cys150Phe)), c.1434G>A (p.(Trp478*)), c.1871C>G (p.(Ser624*)) and c.913del (p.(Leu305fs*1)) in the *XDH* gene and c.1046C>T (p.(Thr349Ileu)) and c.1771C>T (p.(Pro591Ser)) in the *MOCOS* gene. Heterologous protein expression studies revealed that the p.Cys150Phe variant within the Fe/S-I cluster-binding site impairs XDH biogenesis, the p.Thr349Ileu variant in the NifS-like domain of MOCOS affects protein stability and cysteine desulfurase activity, while the p.Pro591Ser and a previously described p.Arg776Cys variant in the C-terminal domain affect Molybdenum cofactor binding. Based on the results of haplotype analyses and historical genealogy findings, the potential dispersion of the identified variants is discussed. As far as we are aware, this is the largest cohort of xanthinuria cases described so far, substantially expanding the repertoire of pathogenic variants, characterizing structurally and functionally essential amino acid residues in the XDH and MOCOS proteins and addressing the population genetic aspects of classical xanthinuria.

## 1. Introduction

Classical xanthinuria is a rare, autosomal recessive metabolic disorder characterized by impaired xanthine dehydrogenase (XDH; EC 1.17.1.4) activity that results in a lack of uric acid (UA) formation and accumulation of its precursors, xanthine (X) and hypoxanthine (HX) [1,2]. Classical xanthinuria occurs worldwide, yet more than two-thirds of cases are reported from the Mediterranean and Middle Eastern countries. The major clinical manifestation of xanthinuria is the formation of xanthine calculi in the urinary tract in about 40% of the affected individuals. Urolithiasis may present as hematuria, crystalluria, recurrent urinary tract infections, renal colic, acute renal failure that may lead to hydronephrosis, chronic renal failure and even death from uremia. Less frequent manifestations are myopathy, arthropathy and duodenal ulcers. Management includes high fluid intake and avoidance of purine and fructose-rich foods [1,2].

Two genetically distinct types of classical xanthinuria can be discerned. Type I xanthinuria (MIM 278300) is caused by variants in the *XDH* gene [3] and results in isolated XDH deficiency. In type II xanthinuria, (MIM 603592) variants in the Molybdenum Cofactor Sulfurase gene (*MOCOS*, alias *HMCS*) cause combined XDH and Aldehyde Oxidase (AO; EC 1.17.3.2) deficiency [4]. The 80.4-kb-long *XDH* gene is located on chromosome 2p23.1 (reference [5], NC_000002.12) and encodes an mRNA of 5.7 kb (NM_000379.4) and a protein of 1333 amino acid residues (NP_000370.2). XDH can be post-translationally converted to an oxidizing form (XO; EC 1.17.3.2). XDH/XO and AO are evolutionarily related proteins that share similar structures and enzymatic activities though with distinct substrate specificities [6,7]. The active form of both enzymes is a homodimer, each subunit containing an N-terminal domain with two (2Fe-2S)-binding centers, a flavin adenine dinucleotide (FAD)-binding domain and a C-terminal molybdopterin cofactor (Moco)-binding domain. The crystal structures of human XDH and AO have been solved, and detailed investigations have been published regarding their structures and mechanisms of action [8,9,10]. The 84.7-kb-long *MOCOS* gene is located on chromosome 18q12.2 (NC_000018.9) and encodes an mRNA of 2.7 kb (NM_017947.1) and a protein of 888 amino acid residues (NP_060417.4). The MOCOS enzyme (EC 2.8.1.9) catalyzes the conversion of the oxo-form to the sulfido-form of Moco that is essential for the activity of both XDH and AO [4]. The MOCOS enzyme is a homodimer, each monomer composed of a Nitrogen fixating Sulfurase-like (NifS-like) N-terminal domain with a pyridoxal-5′-phosphate (PLP)-mediated cysteine desulfurase activity and a C-terminal domain that binds the Moco [11]. To date, there is no quaternary structure of the whole human MOCOS protein available at the Research Collaboratory for Structural Bioinformatics (RCSB) Protein Data Base (PDB) site (rcsb.org).

In the past, we described the molecular basis of classical xanthinuria in five families [12,13,14]. Here, we present clinical, molecular, biochemical and genealogical history data on nine additional Israeli families and two sporadic cases from Germany.

## 2. Materials and Methods

### 2.1. Patients and Control Subjects

Nine families (designated F1–F9 in order of their presentation) and two sporadic cases (G1 and G2) referred to our laboratory between the years 1997 and 2013 and 34 individuals of Yemenite-Jewish extractions consecutively admitted to the Rheumatology Department at the Tel Aviv Medical Center from April 1997 to January 1998 were studied.

### 2.2. Biochemical Tests

Testing for the levels of UA by routine clinical biochemistry analysis and analysis of X and HX, as well as the allopurinol loading test [15] by high-pressure liquid chromatography (HPLC), were performed as previously described [12,13].

### 2.3. DNA Analyses

DNA analysis was performed on 76 samples belonging to the affected families/cases and 34 control samples of Yemenite-Jewish origin. DNA was manually extracted from peripheral blood [12]. Analysis of the Simple Sequence Repeats (SSR; formerly known as microsatellites), Single Strand Conformation Polymorphism (SSCP) analysis using Mutation Detection Enhancement (MDE) gels and Restriction Fragment Length Polymorphism (RFLP) assays, as well as manual and automated Sanger sequencing, were performed as earlier described [12,13]. The recombination map of microsatellite markers on chromosome 18 was taken from Kong et al. [16]. Real-time PCR and a high-resolution melting analysis (HRMA) were performed on a Rotor Gene 6000 instrument (Corbett Research, 14 Hilly Street, Mortlake, NSW, 2137, Australia) and run on software Rotor-Gene, version 1.7.75, according to the manufacturer’s instructions. Syto 9 fluorescent dye (www.thermofisher.com accessed on 4 April 2021) was incorporated into the PCR reaction mix at a final concentration of 1.5 µM.

At the start of this project, the human gene responsible for type II xanthinuria was still unknown; therefore, the following strategy was used for variant detection in the suspected type II xanthinuria patients. The human cDNA database Entrez-Protein (www.ncbi.nlm.nih.gov accessed on 6 June 2020) was queried for the putative bovine MOCOS cDNA (GeneBank ABO36422) [17] using the BLASTN 2.0.2.12 tool, Apr-21-2000 [18]. This search revealed a sequence of a human hepatoma cell line (HepG2)-derived cDNA clone of an unnamed protein product of 888 amino acid residues (AK000740.1, submitted to the DDBJ/EMBL/GenBank databases from the NEDO human cDNA project by Sugano, S. et al. on 5 February 2000). The unfinished High-Throughput Genomic Sequences (HTGS) database (www.ncbi.nlm.nih.gov accessed on 6 June 2020) was then queried with this cDNA sequence, and five high-score genomic clones containing several unordered pieces that mapped to chromosome 18q12 were retrieved: DDBJ: AP001321.1, AP001159.1, AP001158.1, AC023043.2 and AP001155.1. The order of the exons was obtained by alignment with the putative bovine MOCOS gene [17]. Primers were designed, and screening for variants was performed by SSCP analysis of the PCR-amplified exons, including exon-intron boundaries, as previously described [13].

Whole-exome DNA sequencing for the detection of variants in families F4 and F9 was performed by Macrogen (Rockville, MD, USA). Exonic and adjacent intronic regions were enriched from genomic DNA derived from peripheral blood via Agilent SureSelect v5, followed by next-generation sequencing (NGS) on the Illumina HiSeq 4000. Variants were detected, analyzed and reported using the Variantyx Genomic Intelligence platform.

Nomenclature: all variants were described according to the Human Genome Variation Society (HGVS) guidelines (http://varnomen.hgvs.org/bg-material/refseq/ accessed on 4 April 2021), numbering the A of the initiation methionine as +1 in the respective nucleotide reference sequences (NM_) and the corresponding amino acid as +1 in the respective protein reference sequences (NP_).

### 2.4. Bioinformatic and In Silico Analyses

The novelty of the identified variants was verified in the ClinVar database (www.ncbi.nlm.nih.gov>clinvar accessed on 10 May 2021). The pathogenicity of the missense variants was evaluated using the Protein Variation Effect Analyzer Software Tool (PROVEAN, provean.jcvi.org) that provides PROVEAN and SIFT predictions. When the PROVEAN score was equal to or below a predefined threshold (−2.5), the protein variant was predicted to have a “deleterious” effect. If the PROVEAN score was above the threshold, the variant was predicted to have a “neutral” effect. SIFT scores ranged from 0 to 1. The amino acid substitution was predicted as “damaging” if the score was under 0.05 and “tolerated” if the score was over 0.05. In addition, the PolyPhen-2 in silico prediction tool (http://genetics.bwh.harvard.edu/pph2/index.shtml accessed on 10 May 2021) was used. The PolyPhen-2 score ranged from 0.0 (tolerated) to 1.0 (deleterious). Variants with scores ranging 0.0–0.15 were predicted to be benign, variants with scores ranging 0.15–1.0 were possibly damaging and variants with scores ranging 0.85–1.0 were probably damaging.

The degree of conservation and solvent accessibility of the amino acid residues in the MOCOS protein was evaluated as formerly described [13]. The ConSurf server (consurf.tau.ac.il/) bioinformatics tool [19] was used for the analysis of the evolutionary conservation of MOCOS. Multiple Sequence Alignment (MSA) was built using CLUSTALW, and the homologs were collected from UNIREF90.

### 2.5. In Vitro Expression Studies

The generation of *A. thaliana* AtXDH1 expression constructs, expression and purification of AtXDH1 proteins, as well as protein analysis, were described previously by Zarepour et al. [20]. Primers used for introducing the Cys161Ser substitution via PCR mutagenesis were: C161S_for (5′-AAT CTA TGT CGT TGT ACT GGT TAT CGA CCC ATT-3′) and C161S_rev (5′- TCG ATA ACC AGT ACA ACG ACA TAG ATT TCC TGC-3′).

For the expression of wild-type MOCOS-NifS, the first 1524 bp of the full-length human MOCOS open reading frame were cloned into the pQE80 plasmid (Qiagen, Hilden, Germany) according to standard cloning procedures to allow for the heterologous production of a 508-aa MOCOS-NifS protein with an N-terminal fused His_6_ tag for affinity purification. The expression construct for the MOCOS-NifS/Thr349Ile variant was generated accordingly, albeit with a mutated open reading frame generated via PCR mutagenesis using the primers: MOCOS-NifS/T349I_for (5′-GAG AAT ATA AAG CAG CAC ATC TTC ACC TTG GCT CAG-3′) and MOCOS-NifS/T349I_rev (5′-CTG AGC CAA GGT GAA GAT GTG CTG CTT TAT ATT CTC-3′).

The standard expression of MOCOS-NifS and its Thr349Ile variant was performed from the pQE80 plasmid (Qiagen, Hilden, Germany) in freshly transformed *Escherichia coli* BL21 (de3) cells. Cells were grown aerobically in LB medium in the presence of 50-µg/mL ampicillin at 37 °C to an A_600_ = 0.5 before induction. Protein expression was induced with 25-µM isopropyl-β-D-thiogalactopyranoside (IPTG) after the addition of the PLP precursor pyridoxin to the medium (1-mM final concentration) to support the synthesis of the PLP cofactor that is required for MOCOS-NifS functionality. After culturing for 20 h at 26 °C, cells were harvested by centrifugation (12,000× *g*, 4 °C, 5 min) and stored at −70 °C until use. Prior to cell lysis, cells were resuspended and thawed in 5 volumes of lysis buffer (50-mM sodium phosphate buffer, pH 9.3 containing 300-mM sodium chloride, 10% glycerol and 10-mM imidazole). Cell lysis was achieved by passaging through a French pressure cell followed by sonication for 5 min on ice. After centrifugation at 50,000× *g* for 30 min at 4 °C, His_6_-tagged proteins were purified on a nickel-nitrilotriacetic acid superflow matrix (Qiagen, Hilden, Germany) under native conditions at 4 °C in buffers of pH 9.3, according to the manufacturer’s instructions, and eluted in elution buffer (50-mM sodium phosphate, pH 9.3 containing 300-mM sodium chloride, 10% glycerol and 250-mM imidazole). After purification by affinity chromatography, the recombinant NifS-like domain of MOCOS was subjected to PLP quantification and a cysteine desulfurase activity analysis.

To determine the content of protein-bound PLP, PLP-specific absorption at 420 nm was measured in preparations of MOCOS-NifS and its Thr349Ile variant and compared to a PLP standard curve (0–350-µM PLP) as described earlier for the MOCOS-NifS homolog ABA3-NifS from *A. thaliana* [21].

The determination of the cysteine desulfurase activity was performed basically as described previously by Lehrke et al. [21], with the exception that the standard assays contained 25 µg of the respective MOCOS-NifS protein in a total volume of 400 µL of 0.1-M Tris/HCl, pH 9.3 containing 25-mM dithiothreitol.

Cloning, expression and purification of the MOCOS-CT proteins was described earlier by Giles et al. [22], while quantification of Moco and its metal-free precursor molybdopterin (MPT) bound to MOCOS was performed as reported by Wollers et al. [23]. The general principle of the latter assay was that Moco and its precursor MPT, which is chemically and structurally identical except for lacking the Mo atom, were converted into the same stable product, form A dephospho, via oxidation and dephosphorylation. Subsequent to a purification step using QAE chromatography, form A dephospho was detected by HPLC and the obtained peaks compared to peaks of sources of known Moco/MPT contents, such as commercially available xanthine oxidase. The primers used for introducing the Pro591Ser and the Arg776Cys substitutions via PCR mutagenesis were: P591S_for (5′-CAC TAA CCT TTA TCT CTA TTC AAT CAA ATC CTG TGC TGC-3′) and P591S_rev (5′-GCA GCA CAG GAT TTG ATT GAA TAG AGA TAA AGG TTA GTG-3′) and R776C_for (5′-CTC AGC TTG CGT TTT TGT GCC AAT ATT ATT ATC-3′) and R776C_rev (5′-GAT AAT AAT ATT GGC ACA AAA ACG CAA GCT GAG-3′), respectively.

### 2.6. Genealogic Research

The genealogical history of the affected families was studied by the method of “networking” [24] using several sources of information, as described earlier [14].

## 3. Results

### 3.1. Assessment of Xanthinuria and Detection of Variants

Nine Israeli families with 18 affected individuals and two sporadic cases referred from Germany are described. Pedigrees of the studied families, including ethnic origins, the gene in which disease-causing variants were detected and nonrelated congenital disorders, are depicted in Figure 1. The families described in this article belong to ancient, large clans/tribes with branches also living outside of Israel. The results of the historical genealogical investigations of the studied families are presented as Supporting Information.

The clinical features of the affected individuals are summarized in Table 1. The experimental data for the assessment, typing and detection of xanthinuria-causing variants are provided as Supporting Information. Seven variants (four novel) identified in the *XDH* gene and three (two novel) variants in the *MOCOS* gene causing type I and type II xanthinuria, respectively, along with the variants previously reported by us are shown in Table 2. The following is a detailed description of the assessment of xanthinuria and detection of variants in the studied families/cases.

#### 3.1.1. Identification of a Yemenite-Jewish Founder MOCOS Variant (c.1046C>T) in Families F1, F3 and F8

The parents and the oldest sibling in family F1 were born in the city of Ta’izz (Shar’ab District in the Southwest of Yemen). The index case was diagnosed with severe urolithiasis at the age of 40 years and presented with joint pain at the age of 41. This clinical combination of joint pain and urolithiasis raised the diagnostic possibility of gout. However, routine biochemical testing revealed undetectable levels of UA in the serum and urine. Classical xanthinuria was confirmed by high levels of urinary xanthine (X) and hypoxanthine (HX), and three additional affected family members were identified by similar biochemical findings (Appendix A). There was a history of renal stones of unknown composition, and urolithiasis was suspected (based on complaints of occasional painful urination in the absence of urinary tract infection) in several affected and carrier members of this family (Appendix A). In addition, the index case had autoimmune overlap syndrome and other disorders, including metastatic non-small-cell lung cancer, from which he died at the age of 61 years. Detailed clinical information about members of this family are presented in Appendix A. An allopurinol loading test was not available for this family. A homozygosity test approach was taken to type xanthinuria based on the fact that the Yemenite-Jewish population is highly endogamous. A linkage analysis with markers within and around the *XDH* gene showed that the affected individuals were heterozygous for the tested markers (Appendix A), suggesting type II xanthinuria. However, compound heterozygosity and linkage to the *XDH* gene could not be excluded (not shown).

Family F3 was apparently unrelated to family F1, yet originated from the same city, Ta’izz, in Yemen. The proposita, a 69-year-old asymptomatic female, presented with obesity and non-insulin-dependent diabetes mellitus (NIDDM) and was incidentally found to have undetectable levels of UA in the serum and urine. There was no known history of renal stones in this family. The index case reported that her parents were second- or third-grade cousins. Xanthinuria was confirmed by high levels of X and HX in the urine (Appendix A), and type II xanthinuria was diagnosed by an allopurinol loading test (Appendix A).

Family F8 was a third, apparently unrelated, Yemenite-Jewish family, originating from the same area as families F1 and F3. The propositus, a 76-year-old male, was followed for benign prostate hypertrophy, type 2 diabetes mellitus, adenocarcinoma of the colon and metastatic transitional cell carcinoma (TCC) of the bladder, from which he died at the age of 85. There was no evidence for urolithiasis, though near his death, particulate material was noted in the ureters in an ultrasound report. Xanthinuria was suspected because of very low levels of UA in the blood and urine (Appendix A). The parents were not related.

Since, in family F1, the results of a linkage analysis with markers within and around the *XDH* gene were nonconclusive, screening for variants by a SSCP analysis in both the *XDH* gene and a putative *MOCOS* gene (see Methods) was initiated. When type II xanthinuria was diagnosed by an allopurinol loading test in family F3, we assumed that these families may share a common variant in the *MOCOS* gene. Indeed, the SSCP analysis and direct sequencing resulted in the identification of a c.1046C>T (p.(Thr349Ileu)) transition in exon 6 of the putative *MOCOS* gene in both families (Appendix A). In addition, two SNPs, c.1072A>G (rs678560) and c.1164G>A (rs667667), were detected in the same exon. In family F8, the already known Yemenite-Jewish variant was identified by PCR and direct sequencing of exon 6 of the *MOCOS* gene (Appendix A). A survey of 34 Yemenite-Jewish individuals revealed one heterozygous male carrier of the pathogenic c.1046C>T variant. He was heterozygous A/G and homozygous G/G at the c.1072A>G and c.1164G>A SNPs, respectively. This individual was diagnosed with SLE and diffuse large B-cell lymphoma and died at the age of 85 from acute respiratory failure. His father originated from the Shar’ab District of Southwest Yemen, and his mother originated from Aden (Yemen). He was possibly related to family F3, as his father and the mother in family F3 shared the same family name.

The c.1046C>T variant was suspected to be a founder variant, since the three families F1, F3 and F8 and the carrier found in the survey originated from the same region in Yemen. An analysis of two closely linked SNPs, c.1072A/C>G and c.1164G>A, showed that the pathogenic variant co-segregated with the G allele at both polymorphic sites in all three families (Appendix A and Table 3) and was consistent with the same association in the carrier from the survey. Where the G allele at c.1164G>A is a common allele, the reported frequency of the G allele at c.1072A/C>G is 0.059 (ncbi.nlm.nih.gov/snp/rs667667#frequency_tab, accessed on 11 May 2021). A similar frequency (0.071) was found in our survey of 28 Yemenite-Jewish control chromosomes. The association of the *MOCOS* c.1046C>T gene variant with a rare polymorphic allele in all seven, apparently unrelated, affected chromosomes strongly suggests a common founder of this variant. A further haplotype analysis in families F1 and F3, using eight markers covering 14.8 Mbp, showed that the chromosomes bearing the pathogenic variant shared common haplotypes encompassing recombination distances of 2.08–11.67 cM (Table 3).

#### 3.1.2. Identification of a Recurrent XDH Variant (c.2473C>T) in Family F2

Family F2 is a Christian-Arab family from the Lower Galilee. The propositus (V-2), a six-month-old male, presented with renal calculi and splenomegaly of unknown etiology. He was found to have undetectable levels of UA in the serum and strongly decreased levels of UA together with elevated levels of X and HX in the urine (Appendix A). Low UA concentrations in blood and urine were found in two additional family members: V-3 and V-6 (Figure 1). The propositus’ parents were first-degree cousins. Type I xanthinuria was diagnosed by the allopurinol loading test (Appendix A), and the affected siblings, V-2 and V-3, were found to be homozygous for polymorphic markers within and around the *XDH* gene (Table 4 and reference [25] patient G). SSCP screening followed by sequencing revealed a c.2473C>T (p.(Arg825*)) variant in exon 23 of the *XDH* gene that was previously reported in a Czech patient [26]. A high-resolution melting analysis (HRMA) assay was developed for detection of this variant (Appendix A).

An unusual result was noted in the segregation analysis in family F2. Using 14 SSRs covering 10.01 cM revealed that the affected siblings (V-2 and V-3) had identical genotypes, yet the run of homozygosity (ROH) of 2.86 cM (Table 4) was unexpectedly short, considering that the parents were first-degree cousins. For comparison, in an Iranian-Jewish type I xanthinuria patient [12] with apparently unrelated parents but belonging to a highly inbred population, homozygosity was preserved over the whole span of the tested markers (Table 4).

#### 3.1.3. Identification of a Novel XDH (c.913del) Variant in Family F4

Family F4 was an Arab family from a small village residing in Northern Israel. The index case, a 15-month-old male, featured restlessness and macroscopic hematuria. Renal ultrasound revealed bilateral nephrolithiasis, and undetectable UA levels were found in the urine and serum (Appendix A). These clinical and biochemical findings raised the possibility of XDH deficiency. Of note, soon after birth, he was clinically and molecularly diagnosed with prolidase deficiency (reference [27] index case in family 1 and Figure 1). He died of severe lung disease (common in prolidase deficiency) at the age of two years [28]. Additional family members were diagnosed with prolidase deficiency, including his father, who died recently at the age of 52 years from respiratory failure (Figure 1). The parents were first-degree cousins. Type I xanthinuria was suggested by homozygosity mapping with markers in the *XDH* gene (Appendix A), but DNA was not available for further analysis due to the premature death of the index case.

Recently, trio-exome sequencing was performed in this family to search for a diagnosis of his affected brother (V-6), who presented with severe global developmental delay. This analysis revealed that V-6 and his father were homozygous for the genetic variant NM_000379.3:c.913del (p.(Leu305fs*1)) in the *XDH* gene, and his mother was heterozygous. Thus, based on these and former results of the segregation analysis with markers linked to *XDH*, we can assume that the proband (V-5) was also homozygous for this genetic variant. The variant was confirmed by Sanger sequencing (not shown).

#### 3.1.4. Identification of a Novel XDH (c.1871C>G) Variant in Families F5 and F7

Family F5 was a large Bedouin-Arab family living in a town located in Southern Israel. The index case, a 4-year-old boy, presented with kidney stones and undetectable levels of UA in the serum and urine (Appendix A). The UA levels were normal in five additional siblings (not shown). One of his older brothers (IV-3) was born with a small dysplastic right kidney, contributing 24% to total kidney function. The parents showed a high degree of consanguinity. The father had four additional children from a second wife (III-3) who was not related, and individual III-3 had one more offspring (IV-11) from another marriage. Type I xanthinuria was demonstrated by the allopurinol loading test in the index case IV-6 (Appendix A).

Family F7 was also of Bedouin-Arab origin from Southern Israel. Families F5 and F7 reported that their tribes of origin are related. One branch of this extended family (individuals IV-4, IV-5, V-14, V-15, V-16 and V-17) was recently described [14]. The index case (V-15), an asymptomatic young girl, was detected during follow-up for a neurogenic bladder that appeared after the surgical repair of a lipoma in the spinal canal. Two additional affected siblings (V-16 and V-17) were detected by biochemical and molecular testing. An additional asymptomatic, one-year-old girl (V-6) suffering from osteogenesis imperfecta and a mild form of Bartter-like tubulopathy, was suspected to be affected by xanthinuria due to close to zero levels of UA in the blood and urine (Appendix A). A new pediatric case suspected for xanthinuria, followed elsewhere, was recently discovered in the same tribe. The siblings III-5 and III-6 (Figure 1) belonged to a third, historically related Bedouin-Arab tribe. Concentrations of X and HX in the blood/urine, as well as an allopurinol loading test, were not available in this family.

In family F5, screening for variants in the *XDH* gene by SSCP followed by sequencing revealed a c.1871C>G (p.(S624*)) change and a SNP c.1936A>G in exon 18. Specific RFLP (Nla III) and HRMA assays were developed for detection of the c.1871C>G variant (Appendix A). The same variant was identified in family F7. The index case (V-15) and her two affected sibs were compound heterozygotes for c.1871C>G and a c.2164A>T (p.(Lys825*)) variant [14]. Their mother (IV-5) was a carrier of the c.2164A>T variant, and the father (IV-4) was a carrier of the c.1871C>G variant. In another branch of this extended family, a second index case (V-6) was homozygous for the c.1871C>G variant.

#### 3.1.5. Identification of Two XDH Variants in Family F6: One Recurrent (c.141insG) and One (c.913del) Shared with Family F4

Family F6 was of Bedouin-Arab origin living in Northern Israel. The index case (IV-15), a 22-year-old female, presented with low back pain and congenital glaucoma and was found to have very low and undetectable levels of UA in the serum and urine, respectively (Appendix A). She died at the age of 40 years of advanced metastatic breast carcinoma. Further tests of UA in the blood and urine in the family revealed an additional sibling (IV-18) suspected for classical xanthinuria. The parents of the index case were first-degree cousins. Independently, a third case, V-1, an 8-year-old girl belonging to another branch of the family, was suspected for xanthinuria. She presented with abdominal pain, irritation in the urinary tract and microscopic hematuria. The UA concentration in the serum was low and undetectable in the urine. Xanthinuria was confirmed by high levels of X and HX (Appendix A). Type I xanthinuria was suspected in the first index case, IV-15, due to homozygosity for markers in the *XDH* gene (Appendix A) and was confirmed by the allopurinol loading test in the second index case (V-1) (not shown).

An aberrant pattern of SSCP bands in exon 3 of the *XDH* gene in patient IV-15 and the obligatory carriers III-5 and III-6 compared to a control sample, followed by sequencing, revealed a c.141insG (p.(Cys48Lfs*12)) variant and a SNP c.101-35 C>G in IVS 2 (Appendix A). The c.141insG variant was previously reported [29,30]. Another affected child (IV-18) was found to be homozygous for this variant. The second index case (V-1) was heterozygous for the c.141insG variant, and further sequencing revealed heterozygosity for a second (c.913del, p.(Leu305fs*1) variant in exon 11 of the *XDH* gene (Appendix A). HRMA assays were developed for the detection of both pathogenic variants. The c.913del variant can be detected also by an Mwo I RFLP assay (Appendix A).

#### 3.1.6. Identification of a Novel XDH (c.1434G>A) Variant in Family F9

Family F9 was an Arab family living in the north of Israel. The index case, the third child of first-degree cousin parents, presented at the age of 9.6 years with painful hematuria that resolved within a few days. Microscopic hematuria was observed, and the urine culture was negative for bacterial infections. Kidney and urinary tract ultrasonography were normal. At the age of 15 years, he had an episode of *E. Coli* urinary tract infection with epididymitis, which resolved with antibiotic therapy. Otherwise, the boy was asymptomatic, and repeated urinary tract sonography were normal. Repeated blood and 24-h urinary collection tests showed UA levels below the limit of detection, suggesting classical xanthinuria (Appendix A). Homozygosity for a c.1434G>A (p.(Trp478*)) variant was detected by whole-exome sequencing and confirmed by Sanger sequencing in the index case.

#### 3.1.7. Identification of Compound Heterozygosity for MOCOS (c.1088_1089del and c.1771C>T) and XDH (c.449G>T and c.641del) Variants in Cases G1 and G2, Respectively

Both cases G1 and G2 were Caucasian individuals from Germany. Case G1 was a 46-year-old female with very low levels of UA in the serum and urine (Appendix A, 24-h excretion of 3.3 mg), and case G2 was a 61-year-old male with undetectable concentration of UA in the serum and very low levels in the urine (Appendix A, 24-h excretion of 2 mg). Both cases were asymptomatic, with no family history of urolithiasis. Screening of the exons of the *XDH* gene by a SSCP analysis was negative in case G1, and sequencing the exons of the *MOCOS* gene resulted in the identification of heterozygosity for the c.1088_1089 del (p.(Leu363Profs*16) and c.1771C>T (p.(Pro591Ser)) variants (Appendix A). The c.1088_1089del variant was recently submitted to ClinVar by Invitae and Baylor Genetics (https://www.ncbi.nlm.nih.gov/clinvar/variation/VCV001017655.2, accessed on 25 April 2021).

In case G2, screening by a SSCP analysis revealed aberrant bands in exons 6 and 8 of the *XDH* gene, and sequencing revealed compound heterozygosity for the c.449G>T (p.(Cys150Phe)) and c.641del (p.(Pro214Glnfs*4)) variants (Appendix A). The c.641del variant has been reported in patients of Czech and German extraction [26,31,32].

### 3.2. In Silico and Biochemical Characterization of Pathogenic Amino Acid Substitutions in XDH and MOCOS

Among ten distinct variants identified in this study, seven were obviously pathogenic, creating either premature stop codons or frameshifts that encoded truncated proteins (Table 2). These variants were predicted to undergo nonsense-mediated mRNA decay and were not further explored. The pathogenicity of the missense variants identified in the present study, Cys150Phe in XDH and Thr349Ile, Pro591Ser and one previously reported, Arg 776Cys [13], in MOCOS, were further investigated by in silico tools and by in vitro heterologous expression experiments. An analysis of the impact of the variants on the biological functions of the proteins using the PROVEAN Genome Variant tool showed that all four variants were predicted to be deleterious and damaging. The actual scores obtained by PROVEAN for Cys150Phe, Thr349Ile, Pro591Ser and Arg776Cys were −10.02, −3.58, −7.64 and −7.67, respectively, and by SIFT, 0.001, 0.018, 0.000 and 0.000, respectively. The PolyPhen-2 platform predicted that the MOCOS Thr349Ileu variant was possibly damaging, while the MOCOS Pro591Ser, Arg776Cys and XDH Cys150Phe variants were probably damaging.

Cysteine 150 in XDH is part of a well-characterized universally conserved motif of the Fe/S I-binding site of the XDH protein [6], and its conservation was not further inquired. The evolutionary conservation and solvent accessibility of the amino acids Thr349, Pro591 and Arg776 in MOCOS were explored by in silico tools, and the results showed that all three amino acid residues were highly conserved and located within highly conserved amino acid blocks (Figure 2). Where Thr 349 and Pro591 were predicted to be buried and to have a structural role affecting the protein conformation, Arg776 was predicted to be exposed and solvent accessible and, thus, possibly participate in the binding of external molecules.

#### 3.2.1. Cys150Phe in the Fe/S I Cluster Binding Site Affects XDH Biogenesis

Cys150 is one of the four Cys coordinates (Cys113, Cys 116, Cys 148 and Cys150) of the Fe/S I cluster-binding site, a fully conserved motif in eukaryotes and prokaryotes [33]. Since human systems were not available, we took advantage of the already developed systems for cloning and in vitro expression of the plant AtXDH1 [20,34] to study the human Cys150 substitution. It was anticipated that a Cys-to-Phe substitution would grossly affect the overall structure of the XDH protein; therefore, for studying the possible impact of Cys150 on the Fe/S cluster binding and, consequently, on the biogenesis of the mature XDH protein, a mild variant (C161S) equivalent to human Cys150Ser was introduced into the AtXDH1 cDNA of the plant *Arabidopsis thaliana* [34] and expressed in the yeast *Pichia Pastoris*. Surprisingly, an AtXDH1 protein carrying this mild substitution could neither be detected by Coomassie Brilliant Blue staining nor by an immunoblot analysis, whereas the expression of wild-type AtXDH1 yielded a full-length protein of 150 kDa (Figure 3B) with full activity (not shown). Of note, other AtXDH1 variants that carried substitutions either in the Moco-binding site or in the FAD-binding site did not present such a drastic effect on the protein integrity [20].

Unlike the Cys150Phe variant protein, the yields of the Thr349Ile, Pro591Ser and Arg 776Cys MOCOS variant proteins were variable, though sufficient for investigation of the functional role of the respective amino acid residues in the MOCOS proteins.

#### 3.2.2. Thr349Ile in the NifS-Like Domain of MOCOS Impairs Protein Stability and Cysteine Desulfurase Activity

In order to study the impact of the Thr349Ile substitution on the function of the MOCOS protein, the corresponding variant was introduced into the cDNA of the NifS-like domain of human *MOCOS* and expressed in *E. coli*. The Thr349Ile substitution resulted in greatly reduced yields of the variant protein compared to the control (Appendix A). An analysis of small amounts of the full-length variant protein that could be isolated exhibited a reduced capacity to bind the NifS-typical PLP cofactor, accompanied by a dramatic drop of the PLP-dependent cysteine desulfurase activity (Figure 3C).

#### 3.2.3. Pro591Ser and Arg 776Cys in the C-Terminal Domain of MOCOS Effect Moco Binding

Both residues, Pro591 and Arg776, are highly conserved among the MOCOS proteins of various origins and in the human MTARC1 protein (Figure 2C). Moreover, the crystal structure of the human MTARC1 protein [35] shows striking similarities to the C-terminal domain of MOCOS proteins and, likewise, resembles a Moco-binding protein. Upon the introduction of either a Pro591Ser substitution or a Arg776Cys substitution into the C-terminal domain of MOCOS and the heterologous expression in *E. coli*, the capability of both MOCOS variants to bind Moco/MPT was dramatically reduced, with the Pro591Ser variant showing a binding capacity of approximately 24% and the Arg776Cys variant about 6%, as compared to the wild-type protein (Figure 3D).

## 4. Discussion

We described nine Israeli families and two isolated cases from Germany from clinical, molecular, biochemical and population genetics perspectives. We report 10 (six novel) variants, biochemically characterized *XDH* and *MOCOS* missense variants and disclosed a Yemenite-Jewish founder effect and historical genealogy data suggesting the dispersion of some variants in the Arab world.

### 4.1. Structural/Functional Significance of Pathogenic Amino Acid Substitutions in XDH and MOCOS

Among the fourteen distinct variants identified in our laboratory (Table 2), of special interest were the variants predicting amino acid substitutions in XDH (Cys150Phe) and MOCOS (Thr349Ileu, Pro591Ser and Arg776Cys). All four variants were predicted by in silico platforms to be deleterious (PROTEAN) and damaging (SIFT and PolyPhen-2), and the respective amino acid residues were found to belong to highly conserved amino acid motifs (Figure 2). Further in vitro expression studies (Figure 3) elucidated the biochemical basis of the pathogenicity of the missense variants. In the case of the type I xanthinuria-causing variant that led to a Cys-to-Phe substitution at position 150, the *Arabidopsis thaliana* XDH isoform 1 (AtXDH1) was used as a model to study the effect of a substitution of Cys150 in human XDH. Importantly, Cys150 represents the C-terminal Cys residue within a -Cys-Xaa2-Cys-//-Cys-Xaa1-Cys- motif of the Fe/S cluster-binding domain of XDH. Cys150 is predicted to ligate one iron atom of (2Fe-2S) cluster I, as was shown for the corresponding Cys150 in the crystal structure of bovine xanthine dehydrogenase/xanthine oxidase (XDH/XO) [35]. While Cys150 has not been characterized previously in eukaryotic XDH enzymes, its equivalent at position Cys136 has been substituted for Ala in *Rhodobacter capsulatus* XDH (along with Cys134) and has been the subject of biochemical and spectroscopic analyses [36]. Moreover, Cys115 and Cys51 in the Fe/S clusters I and II, respectively, of rat XDH have been intensively characterized by substituting these residues for Ser in recombinant enzymes [37]. For each of these Cys variants of rat or bacterial XDH enzymes, the resulting proteins were found to be predominantly in unstable monomeric forms, lacking the Moco and with no or negligible activity. Thus, it was concluded that these Cys residues are not only important for Fe/S cluster binding but, also, for proper protein folding, dimer assembly and the insertion of Moco [36,37]. This is well in line with our observations of the AtXDH1/Cys161Ser variant, which, likewise, was demonstrated to be highly unstable, as no full-length protein could be obtained after expression in the methylotrophic yeast *Pichia pastoris* (Figure 3B), indicating that Cys161 in AtXDH1 and its counterpart Cys150 in human XDH is crucial for protein stability. Taken together, the findings of the present work, along with those of Iwasaki et al. [37] and Schumann et al. [38], suggest that the impairment of the XDH function in case G2 and the manifestation of type I xanthinuria is due to a severe disruption of the process of biogenesis of XDH, originally caused by the c.449G>T variant.

MOCOS proteins basically consist of two domains: a N-terminal NifS-like domain and a C-terminal Moco-binding domain [9]. The NifS-like domain catalyzes the mobilization of sulfur from L-cysteine in a PLP-dependent manner via its so-called L-cysteine desulfurase activity, with the liberated sulfur serving as a substrate for sulfuration of the C-terminally bound Moco. The sulfurated form of the latter is required for full maturation of Mo-hydroxylase enzymes of the xanthine oxidase (XO) family, such as XDH/XO and aldehyde oxidase (AO). It is thus not surprising that variants either in the N-terminal NifS-like domain or in the C-terminal Moco-binding domain of MOCOS proteins have been found to affect the activation of both XDH/XO and AO. However, even though several variants have been reported to be linked with type II xanthinuria [4,13,39,40], no direct correlation has been provided so far between the variant and the precise biochemical effects on MOCOS enzymes. Yet, our characterization of the recombinant MOCOS variant Thr349Ile showed that the type II-linked c.1046 C>T variant resulted in a reduced yield, reduced binding of the PLP cofactor and absence of cysteine desulfurase activity (Figure 3C). The corresponding Arabidopsis variant, ABA3-NifS T354I, was actually found to be very similar to the human MOCOS-NifS T349I variant: wild-type-like distribution of bands on SDS gels, approximately 40% PLP saturation and 10–15% cysteine desulfurase activity (Rump and Bittner, unpublished results). It is recognized that Thr349 itself is conserved among MOCOS proteins (Figure 2B), albeit not as part of the prominent motif around Lys264, which its counterpart Lys 271 in the plant MOCOS protein ABA3 has been demonstrated to be crucial for PLP-binding [41]. Rather, in a theoretical structure of MOCOS-NifS deduced from the available 3D structure of the *E. coli* cysteine desulfurase SufS (PDB no: 1JF9 and 1C0N), Thr349 is located within an α-helix without direct contact with the PLP cofactor. It is thus concluded that the substitution of Thr349 by Ile affects the protein conformation in a way that the protein is less stable compared to the wild-type protein, and the proper insertion and/or stabilization of PLP within the MOCOS backbone is impaired, which, in consequence, is accompanied by a loss of catalytic activity.

Both Pro591 and Arg776 are located within highly conserved motifs of the C-terminal Moco-binding domain of MOCOS (Figure 2B,C). Interestingly, Arg776 in human MOCOS corresponds to Arg723 in the plant MOCOS protein ABA3, which is affected by a mutation in the *sir3-3* mutant [42] that presents negligible activities of its target enzymes XDH and AO [23]. Heterologous expression of the ABA3 protein reflecting the *sir3-3* mutation, i.e., ABA3/Arg723Lys, has shown that the incapacity to activate XDH and AO derives from a reduced ability of the ABA3 variant to bind Moco [23]. In fact, a strong effect on the capability of human MOCOS to bind Moco is likewise observed for the Arg776Cys variant studied in this work, thereby proving the general importance of Arg776 (and Arg723 in ABA3, respectively), for the binding of Moco in MOCOS enzymes. This is further substantiated by the structural analysis of the human MTARC1 (mitochondrial amidoxime-reducing component) enzyme, which shows significant sequence similarities to the C-terminal domain of MOCOS proteins and is a member of the Moco sulfurase C-terminal (MOSC) domain family [35]. MTARC1 catalyzes the NADH-dependent N-reductive cleavage of N-hydroxylated substrates in concert with the electron transport proteins cytochrome b_5_ and cytochrome b_5_ reductase [43].Within a conserved FR(A/P)N motif (Figure 2C), Arg238 of human MTARC1, which corresponds to MOCOS Arg776 (and Arg723 of ABA3, respectively), was found to be the ligand to the pterin ring system and the phosphate moiety of bound Moco [35]. It is thus assumed that Arg776 in human MOCOS has a similar function with its malfunction or substitution resulting in an impaired Moco-binding capacity, as observed for the recombinant Arg776Cys protein. Besides the Arg776Cys substitution, the human MTARC1 structure also provides an explanation for the Pro591Ser substitution, which was likewise shown to go along with the reduced Moco-binding capacity. Like Pro591 in MOCOS, the corresponding residue in human MTARC1, Pro65, is part of a YP(I/V)KSC motif (Figure 2B), for which a function related to Moco binding could be ascribed. Specifically, Lys67 (K67) and Ser68 (S68) were shown to coordinate the phosphate moiety in human MTARC1 [35], so that a substitution of Pro65 in the near proximity of these residues is likely to affect the bonds between Lys67 and/or Ser68 and the Moco phosphate moiety. In relation to MOCOS, it is therefore very likely that the substitution of Pro591 affects the properties of Lys593 and Ser594 in terms of binding the phosphate moiety of Moco, which is indeed exactly what has been found by the characterization of the recombinant Pro591Ser variant. Thus, with the biochemical characterization of the recombinant MOCOS proteins carrying amino acid substitutions, as found in families F1, F3 and F8 (Thr349I), case G1 (Pro591Ser) and the case described by Peretz et al. [13], Arg776Cys in the present work for the first time provides evidence for the biochemical and functional basis of type II xanthinuria.

### 4.2. Potential Dispersion of the Identified Variants in Specific Populations

The families studied in Israel, with one exception (reference [13] family ZO), belong to ethnic groups (Iranian-Jewish; Yemenite-Jewish and Bedouin, Muslim or Christian Arabs) characterized by high rates of consanguineous and endogamous marriages. Among the 17 couples with affected offspring studied by us, 70% were consanguineous and 18% endogamous. Due to this special marriage pattern, most affected offspring (81%, N = 27) were homozygous for the pathogenic variants, as opposed to cases G1 and G2 from Germany, who were both heterozygous. This special make-up of the population of origin of the studied families also makes it plausible that the variants identified in more than one family (Table 1) stem from a common ancestor. For the MOCOS c.1046C>T variant found in the Yemenite-Jewish families F1, F3 and F8, a founder effect was demonstrated by haplotype analyses (Table 2). The allelic overlaps on the affected chromosomes in families F1 and F3 (recombination distances of 2.08, 4.85 and 11.37cM) suggest a less ancient common ancestor than that of the Arab-Turkmen XDH variant where a shared haplotype of 0.34 cM predicted 179 (95% credible limit 70) generations to the MRCA [14]. Regarding the XDH variant c.1871C>G identified in families F5 and F7, these families are known to belong to frequently intermarrying, historically related tribes; therefore, a haplotype analysis was not further pursued. Concerning another XDH variant, c.913del identified in families F4 and F6, genealogical research showed possible historical relations between these families; however, a demonstration of common ancestry by a haplotype analysis was not feasible due to the recent identification of the variant in family F4. Another possibly old XDH variant, c.2473C>T, was identified in family F2 by homozygosity testing. The ROH in two affected siblings was surprisingly short, 2.86 cM (Table 3), considering that the parents were first-degree cousins (Figure 1). For comparison, the ROH was at least 10.01 cM long in an Iranian-Jewish patient (Table 3) and at least 5.62 cM long in siblings from Arab and Turkmen families [14]. One explanation for this observation might be, though with very low probability, that recombination occurred at two close sites around the pathogenic variant in two generations. A more probable explanation is that the MRCA of the variant was a distant relative and that this variant was established in the extended family many generations ago. Further work is needed to clarify this issue. Of note, this variant located at a CpG mutation hot spot in the XDH gene was previously reported in a compound heterozygous Czech xanthinuria case [26]. Under the assumption that at least some of the variants identified in the Israeli-Arab families might be old and the fact that these families came from and still have relatives living in diverse Arab countries, it is likely that the identified variants are dispersed not only in Israel but also elsewhere in the Arab world.

### 4.3. Clinical Aspects

Classical xanthinuria is a rare autosomal recessive disorder apparently more frequent among inbred populations, yet probably underreported in general [2,44]. Xanthine stones are very rare. An analysis of more than 10,000 stones in the adult US population did not reveal a single case of a xanthine-composed stone [45]. In a Morocco analysis of stones collected over a period of 17 years, it showed that stones in children are rare compared to adults (432/9400, 4.6%), and xanthine stones in children accounted for only 0.5% of cases [46]. In Tunisia, over a period of 10 years, less than 3% of the infants were referred for a renal stone [47], and, in Kuwaiti children with urolithiasis, four out of 31 cases (12.9%) were affected by xanthinuria [48].

Based on more than 150 reported cases, symptoms that could be attributed directly to xanthinuria have been reported in around 40% of patients, with no distinction between the clinical manifestations of type I and type II patients [2]. Our clinical findings were similar (Table 1 and references [12,13,14]). Symptoms of xanthine urolithiasis may occur as early as in the neonatal period [49] or in childhood, as described by us (index cases in families F2 and F5 in Table 1 and references [13,14]) and others [29,30,50] or may present later in life, as in family F1 (Table 1) and other reports [51]. In an older review of more than 100 cases, the proportion of children vs. adults presenting with symptomatic xanthinuria was around 50% [1]. Publication bias may exist, as the young-age cases are more dramatic, are anticipated to occur in a genetic background and, thus, are thoroughly investigated. In our series, the proportion of symptomatic children was higher than in adults. Eight (four type I and four type II) out of 17 children and two young adults (47%) presented with symptoms that could be attributed to xanthinuria (Table 1 and references [13,14]). Severe congenital disorders present in these families (Figure 1) that needed repeated biochemical screening facilitated the detection of additional asymptomatic cases through hypouricemia due to the awareness of the treating pediatricians for rare inherited diseases.

In contrast, it seems that most reported adult patients affected by xanthinuria were asymptomatic and fortuitously detected [3,4,31,32,39,40]. In our series, among the 11 adult subjects (three type I and eight type II), none presented with xanthinuria-related complaints, yet three (27%) were found retrospectively to have suffered of symptoms of urolithiasis (Table 1). The identification of xanthinuria in family F1 demonstrated how the diagnosis of xanthinuria can be missed in adult patients even when presenting with urolithiasis. The index case II-8 approached the Rheumatology Department at the age of 41 years because of joint pains. A review of his medical history revealed that he was diagnosed earlier with a kidney stone for which he underwent extracorporeal shock wave lithotripsy. Since urolithiasis in adults is not such a rare event, the stone composition or its cause were not further investigated. Suspicion of gout disease and testing for UA at the Rheumatology Department finally led to the diagnosis of xanthinuria. Moreover, a history of nephrolithiasis was retrospectively revealed in several homozygous and heterozygous family members in this family (Appendix A), as well as in potential (not tested) carrier relatives in other families followed up on by us [12,13]. Similar findings were previously reported [50]. It seems that the carrier status of xanthinuria and other genetic and environmental factors may play a role in an enhanced susceptibility for urolithiasis in the affected families. Preventive therapies for stone recurrence in xanthinuria include copious drinking and the avoidance of purine and fructose-rich foods [1,2]. Urolithiasis did not reoccur in any of our patients. There was no need for a prenatal diagnosis of classical xanthinuria [1].

In view of the diverse pathophysiological roles suggested for the enzymes involved in xanthinuria, the question arises whether the enzymatic deficiencies induce clinically significant perturbances. The proposed pathophysiological role of XDH and its oxidase form XO is highly complex [52,53,54]. On the one hand, UA, the final product of XDH, is a reducing agent, and it was proposed to play a protective role in humans; on the other hand, XDH/XO is a major source of reactive oxygen species (ROSs) that, in excess, are considered detrimental to health. Elevated levels of XDH/XO in the serum were found in pathological situations, such as rheumatic and autoimmune diseases, schizophrenia, type 2 diabetes, hypertension, dyslipidemia and cancer, and it was proposed that proinflammatory and prothrombotic activities of endothelial cells induced by high levels of circulating XDH/XO may contribute to the pathogenesis of atherosclerosis [53]. The physiological role of AO in humans is even more obscure [7]. There is no known human disease associated with isolated AO deficiency [52]. Two examples of experimental findings linked to AO deficiency might be of clinical interest. Knockout mice of one of the isoforms of AO (*Aox4*-/-) were characterized, among other features, by a resistance to diet-induced obesity and hepatic steatosis [7], while an accumulation of N-methyl-nicotinamide in type II xanthinuria was suggested to increase the risk for type 2 diabetes [55,56]. The downregulation and diminished activity of MOCOS were recently associated with autism and other neuropsychiatric disorders [57,58,59,60], and at least one case of type II xanthinuria associated with autism was reported [61]. How can these observations be reconciled with the clinical findings in our series of adult cases? Among the adult type I xanthinuria cases, case G2 was apparently healthy at the age of 61 and was lost to follow-up. Individual IV-7 in family F4 was affected by another autosomal-recessive disease, prolidase deficiency, that caused his death due to respiratory failure at the age of 52 years. The Iranian-Jewish patient described by Levartovsky et al. [12] was diagnosed with hypertension at the age of 32 years and, at presentation (age 60 years), complained of muscle and joint pains and suffered from heart disease. She died at the age of 65 years. In the adult type II cases studied by us, several autoimmune disorders, hypertension, obesity, type 1 and type 2 diabetes, NAFLD, CVA and various malignancies were observed (Table 1 and reference [13]). Our data showed that pathologies suggested to be linked to high levels of XDH/XO activity were nevertheless present in both type I and type II patients who were deficient for this enzyme. In addition, the type II subjects (II-7 in family F1, II-2 in family F3 and the index case in family ZO [13] presented with obesity and/or NAFLD, unlike the AO knockout mice, which were resistant to these pathologies. On the other hand, the accumulation of N-methyl-nicotinamide due to a lack of AO activity in the affected individuals in families F3 and F8 may have contributed to the development of type 2 diabetes. No symptoms attributed to MOCOS deficiency were observed in the present studied cohort. Our clinical observations suggest that, in humans, the effect of XDH, AO or MOCOS deficiencies, whether exerting beneficial or detrimental effects, is not prominent. However, they could have a role as modifier genes among other genetic and environmental factors involved in the causation of the final phenotype, especially in consanguineous kindreds with multiple autosomal-recessive disorders.

An important health aspect of the enzymatic deficiencies in xanthinuria is their involvement in drug metabolisms [62,63]. One recent example is the clinical trial of the Favipiravir and Hydroxychloroquine combination treatment of COVID-19, from which subjects affected by xanthinuria were excluded [64]. It was shown that not only a full but, also, a partial deficiency of the enzymes involved may affect the optimal drug dosage [40,65,66,67]. From this point of view, it is of importance to detect not only homozygous or compound heterozygotes but, also, heterozygous carriers of deleterious variants. Since biochemical tests (levels of UA in the blood or urine, Peretz et al., unpublished results, or XDH activity in the serum [26]) are not useful for detecting carriers, the availability of molecular information and affordable assays for the detection of specific variants, especially in large kindreds, has important clinical significance. Ours and others’ works illustrate the technological advancement in the typing and molecular diagnosis of xanthinuria, from labor-intensive manual methodologies to highly automated omic approaches [12,15,55,68,69,70]. Still, problems may be encountered in special cases like the identification of intronic variants [70,71] or very large heterozygous deletions [72]. As whole-genome sequencing is becoming clinically feasible and affordable, it is expected that these problems will be overcome.

## 5. Conclusions

As far as we are aware, we described the largest series of xanthinuria cases reported so far. We reported six novel pathogenic variants, and for the first time, we characterized MOCOS variants underlying type II xanthinuria by in vitro expression studies. We disclosed a new Yemenite-Jewish founder variant and presented molecular and historical data suggesting the prevalence of some variants not only in Israel but, also, elsewhere in the Arab world.

## Figures and Tables

**Figure 1 biomedicines-09-00788-f001:**
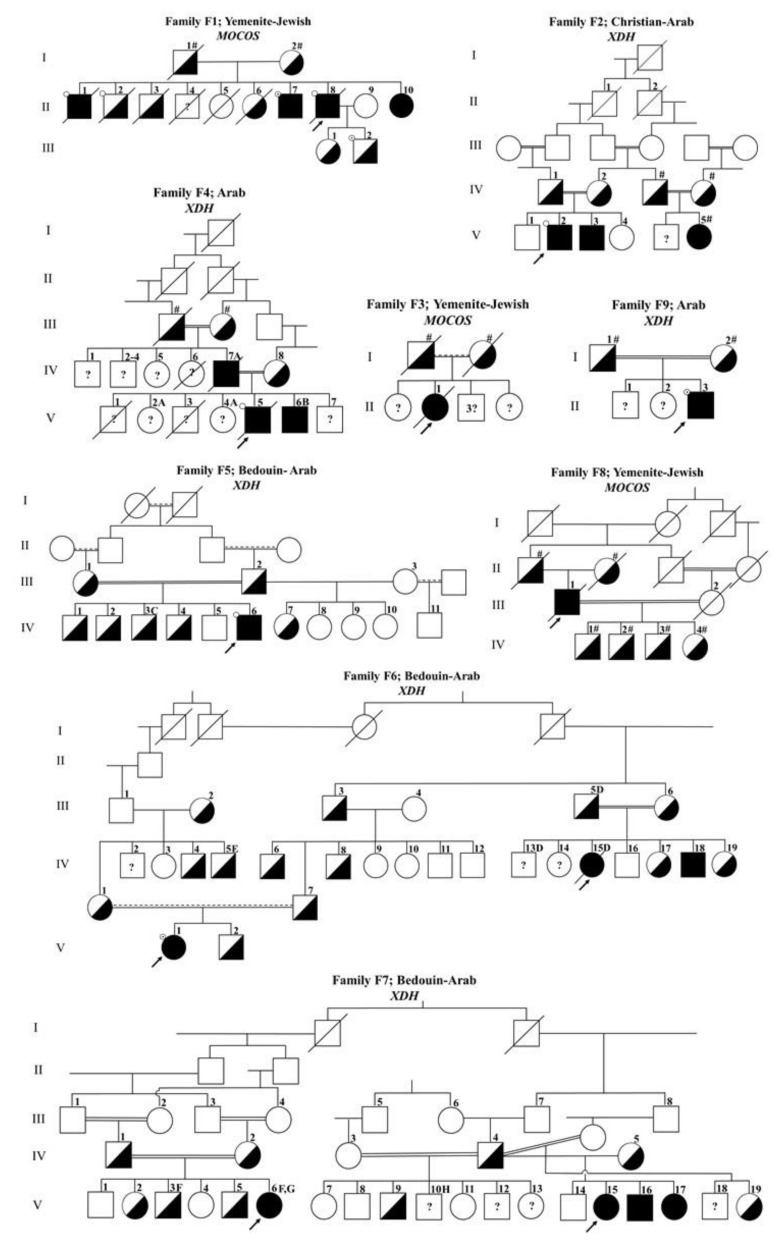
Pedigrees of nine Israeli families affected by classical xanthinuria. Symbols: filled circle/rectangle—affected female/male, half-filled circle/rectangle—obligatory/identified carrier of disease-causing variant, arrows point to the index case, ?—no samples available for analysis, #—not genotyped obligatory carrier/affected and a small circle without or with a dot on the upper-left side of the symbol represents a kidney stone or suspected urolithiasis, respectively. Congenital disorders observed in family members: A—prolidase deficiency, B—severe global developmental delay, C—small dysplastic kidney with reduced function, D—congenital glaucoma, E—spherocytosis, F—Bartter-like tubulopathy, G—Osteogenesis imperfecta and H—spina bifida.

**Figure 2 biomedicines-09-00788-f002:**
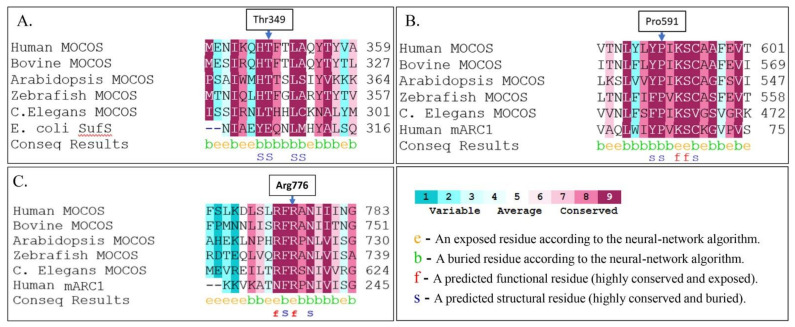
Evolutionary conservation of the substituted amino acid residues Thr349Ileu (panel **A**), Pro591Ser (panel **B**) and Arg776Cys (panel **C**) in the MOCOS protein identified in patients affected by type II xanthinuria. Upon sequence comparisons across 314 PSI-BLAST hits, 294 of them were identified as unique, and conservation calculations were performed on the 150 sequences closest to human MOCOS (NP_060417.4): Bovine MOCOS NP_776506.1, Arabidopsis MOCOS Q9C5 × 8.1, Zebrafish MOCOS A2VD33, *C. Elegans* MOCOS Q21657, *E. Coli* SufS NP_416195.1 and human MTARC1 NP_073583.1. Human MTARC, i.e., human mitochondrial amidoxime reducing component, presents significant similarities along its entire length (excluding its N-terminal mitochondrial targeting sequence) to the C-terminal domain of MOCOS. Both MTARC and MOCOS proteins are members of the MOSC (Molybdenum Cofactor Sulfurase_C-terminal) domain family, with the recently solved three-dimensional structure of human MTARC1 allowing for structural comparisons between human MTARC1 and MOCOS proteins. Note that all three amino acid residues, Thr349, Pro591 and Arg776, substituted in the variant proteins are highly conserved and located within highly conserved amino acid blocks. While Thr349 and Pro591 are predicted to be buried amino acid residues, Arg776 is predicted to be exposed and solvent-accessible.

**Figure 3 biomedicines-09-00788-f003:**
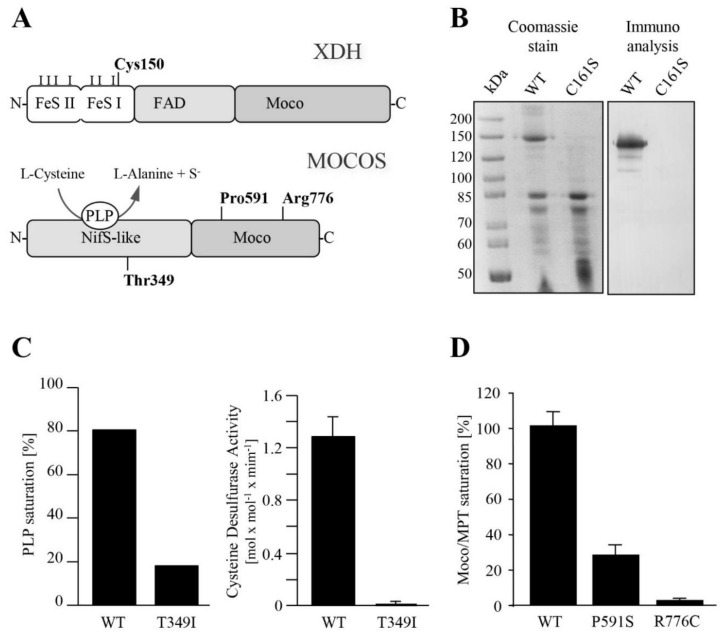
Biochemical characterization of xanthinuria-linked patient variants in XDH and MOCOS. (**A**) Schematic presentation of human XDH (top) and MOCOS (bottom) protein monomers with their respective functional domains and the residues affected by the patient variants (Fe/S, iron sulfur clusters I and II; FAD, flavin adenine dinucleotide; Moco, molybdenum cofactor; PLP, pyridoxal-5-phosphate). Bars on the Fe/S domain of XDH indicate cysteine residues involved in Fe/S cluster binding. (**B**) Expression analysis of recombinant wild-type Arabidopsis XDH isoform 1 (AtXDH1, WT) and its Cys161Ser variant as expressed in *Pichia pastoris* and purified via Ni-NTA affinity chromatography, with Cys161 in AtXDH1 reflecting residue Cys150 in XDH of xanthinuria type I patients. Left panel: recombinant AtXDH1 proteins after electrophoresis on 7.5% SDS polyacrylamide gels and staining with Coomassie Brilliant Blue. Right panel: immunoblot analysis of recombinant AtXDH1 proteins using His_6_-specific antibodies. The dominant band at approx. 150 kDa corresponds to XDH full-length monomers, while bands below this represent XDH degradation products. (**C**) PLP saturation and cysteine desulfurase activity of the wild-type NifS-like domain of human MOCOS and its Thr349Ile variant as heterologously expressed in *E. coli* and purified via Ni-NTA affinity chromatography. For the PLP saturation, the results of a single “good” protein preparation experiment is shown, where PLP binding was measurable in contrast to other preparations in which PLP binding was beyond the limit of detection. This specific protein preparation experiment is included in the results shown on the right side of the panel to emphasize that, even when certain PLP binding could be demonstrated in the variant protein, the cysteine desulfurase activity was still close to zero. (**D**) Moco/MPT contents of the wild-type C-terminal domain of HMCS/MOCOS and its Pro591Ser and Arg776Cys variants, as heterologously expressed in *E. coli* and purified via Ni-NTA affinity chromatography. Moco/MPT contents were quantified by the HPLC-based detection of their common oxidation product, Form A dephospho. It should be emphasized that the amount of the control and variant proteins (56 kDa and 38 kDa bands, respectively; Appendix A) were evaluated by densitometry and adjusted accordingly in the assays.

**Table 1 biomedicines-09-00788-t001:** Clinical findings in 20 subjects affected by classical xanthinuria.

Family/Case	Type	Case	Sex/Age ^1^ (y)	Clinical Manifestations ^2^	
				Xanthinuria	Other
F1	II	II-1	M 64 80 *	UL	RA, vitiligo, DM**, HTN, tinnitus, dementia, CVA, leukoaraiosis
		II-7	M 49	(UL)	Sjogren’s syndrome, myeloma, HTN, NAFLD
		II-8	M 41 61 *	UL	autoimmune overlap syndrome ^3^, NSCLC, DM1, dyslipidemia, cholecystitis, osteoporosis
		II-10	F 44	-	breast cancer, Hashimoto
F2	I	V-2	M 0.5	UL	splenomegaly
		V-3	M 5	-	-
		V-5	F 5	-	-
F3	II	II-1	F 69 72 *	-	obesity, DM2
F4	I	IV-7	M 52 52 *	-	prolidase deficiency
		V-5	M 1.25 2.5 *	UL	prolidase deficiency
		V-6	M 1.5	-	-
F5	I	IV-6	M 4	UL	-
F6	I	IV-15	F 22 40 *	low-back pain	congenital glaucoma, breast cancer
		IV-18	M 24	-	-
		V-1	F 8	-	-
F7	I	V-6	F 1	-	osteogenesis imperfecta, mild Bartter-like tubulopathy
F8	II	III-1	M 70 85 *	-	DM2, TCC of bladder, colon cancer, TIA, ESRD
F9	I	II-3	M 9.6	UL	-
G1	II		F 46	-	lost to follow-up
G2	I		M 61	-	lost to follow-up

^1^ At presentation, * at death, ^2^ including follow-up, ^3^ RA—Sjogren’s syndrome-lupus-scleroderma-myositis overlaps with UIP (Usual Interstitial Pneumonia), UL—urolithiasis, (UL)—Suspected urolithiasis, RA—rheumatoid arthritis, DM1/2—diabetes mellitus type 1/2, DM**—diabetic patients who were switched from oral hypoglycemic agents to insulin therapy due to inefficacy, HTN—hypertension, CVA—cerebrovascular accident, NAFLD—nonalcoholic fatty liver disease, NSCLC—non-small cell lung cancer, TCC—transitional cell carcinoma, TIA—transient ischemic attack and ESRD—end-stage renal disease.

**Table 2 biomedicines-09-00788-t002:** Variants causing classical xanthinuria identified in patients referred to the Tel Aviv Medical Center between the years 1997 and 2013.

Gene	Exon	Variant ^1^	Protein	Family/Case	Reference
*XDH*, NM_000379.4 NP_000370.2	3	c.141insG	p.(C48Lfs*12)	F6	29, 30
	6	**c.449G>T**	**p.(C150F)**	G2	-
	8	c.641del	p.(P214Qfs*4)	G2	26, 31, 32
	11	**c.913del**	**p.(L305*1)**	F6, F4	-
	15	**c.1434G>A**	**p.(W478*)**	F9	-
	16	c.1658insC *	p.(A556Sfs*67)		12
	18	**c.1871C>G**	**p.(S624*1)**	F5, F7	-
	20	c.2164A>T *	p.(K722*)		14, 50
	23	c.2473C>T	p.(R825*)	F2	24
*MOCOS* NM_017947.1 NP_060417.4	6	c.1037insA *	p.(Q347Afs*33)		13
	6	**c.1046C>T**	**p.(T349I)**	F1, F3, F8	-
	6	c.1088_1089del	p.(L363Pfs*16)	G1	CinVar ID 1017655 dbSNP-rs761752580
	8	**c.1771C>T**	**p.(P591S)**	G1	-
	13	c.2326C>T *	p.(R776C)		13

^1^ Novel variants are highlighted by bold letters; * previously reported by us.

**Table 3 biomedicines-09-00788-t003:** Haplotype analysis of the polymorphic markers linked to the *MOCOS* c.1046C>T variant in three Yemenite-Jewish families.

	Chromosomal Position		Family		
Marker	bp *	cm **	F1	F3	F8
D18S1104	21,579,772	42.72	4/6	2/5	-
D18S1107	24,548,186	45.83	2/**3**	**3**/1	-
D18S56	30,715,573	52.48 ***	**2 2**	**2**/4	-
D18S456	33,582,802	55.42	**2 2**	**2 2**	-
**c.1046C>T**	36,205,105	57.33 ***	**T T**	**T T**	**T T**
c.1072 A >G ^#^	36,205,131	idem	**G G**	**G G**	**G G**
c.1164 G>A ^##^	36,205,223	idem	**G G**	**G G**	**G G**
(GT)_n_ -IVS12	36,254,460	idem	**5 5**	**5 5**	-
D18S1093	36,439,598	57.50	4/**1**	**1 1**	-

***** GRCh38.p13 primary assembly, annotation release 109 (https://www.ncbi.nlm.nih.gov/ accessed on 10 September 2020); ** reference [16]; *** estimated by linear interpolation and ^#^ rs678560 and ^##^ rs667667 (NCBI dbSNP database). The pathogenic variant and shared alleles are highlighted in bold.

**Table 4 biomedicines-09-00788-t004:** Haplotype analysis of the markers around the *XDH* gene variants c.2473C>T and c.1658insC.

Marker	Chromosomal Position		Case	
	bp *	cM **	F2 V-2/V-3	IJ
D2S165	28,380,642	50.07 ***	4 7	**6 6**
D2S170	28,991,010	50.94	3 5	**4 4**
D2S146	29,369,271	51.48	6 6	**1 1**
D2S375	30,737,968	-	3 2	**3 3**
D2S400	30,920,642	54.55	3 1	**5 5**
D2S2255	31,000,369	54.66	2 5	**2 2**
D2S2283	31,211,672	55.03	**1 1**	**3 3**
D2S352	31,278,307	55.21	**5 5**	**5 5**
D2S2203-*XDH*	31,384,160	55.37	**3 3**	**3 3**
D2S2351	31,861,912	-	**3 3**	**2 2**
D2S2325	32,930,450	-	**5 5**	**2 2**
D2S2347	33,151,729	56.05	**4 4**	**3 3**
D2S367	34,216,223	57.89	**3 3**	**7 7**
D2S2374	35,484,977	60.08	5 3	**2 2**

* GRCh38.p13 primary assembly annotation release 109 (https://www.ncbi.nlm.nih.gov/ accessed on 10 September 2020); ** reference [16]; *** estimated by linear interpolation and the run of homozygosity is highlighted in bold. IJ—Iranian-Jewish patient in reference [12].

## Data Availability

Not applicable.

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
