# Peer review of "Classical Xanthinuria in Nine Israeli Families and Two Isolated Cases from Germany: Molecular, Biochemical and Population Genetics Aspects"

_biomedicines, 2021, doi:10.3390/biomedicines9070788_

Round 1
Reviewer 1 Report
Comments for Authors
Summary
The manuscript by H Peretz and colleagues describes a series of nine families from Israel (of Yemenite Jewish and Arab ancestry) and two additional individuals from Germany that have classical xanthinuria. One of the nine families was found to correspond to a new branch of one of five original families that that authors had previously reported. Xanthinuria is a rare condition that results from loss of function in either of two genes, XDH (type I) or MOCOS (type II). A total of ten candidate causal variants were identified yielding 14 candidate causal variants overall, five in MOCOS and nine in XDH. Of these, six had not been previously described, including one single nucleotide deletion leading to frameshift and five nucleotide changes. Two of the five changes predict premature stop codons and three involve missense substitutions.
The presented work provides detailed family histories with extensive clinical information.
In vitro investigations are described to support that the three new missense variants (C150F in XDH and T349I and P591S in MOCOS) and an earlier identified missense variant (R776C in MOCOS) are deleterious.
Concerns/Comments
- The clinical descriptions are lengthy and may benefit from more effective use of tabular formatting, at least for anticipated complications (issues and symptoms related to xanthinuria) and for some of the clinical details that are not suspected to be directly related (based on underlying mechanism or observed family segregation information).
- The details of the ancestry of the families with xanthinuria are interesting, however so much information is given that families themselves are identifiable. This would appear to go beyond standard consent given for inclusion in research studies. Some of the background details should not be included to protect the identity of the families. In addition, Section 3.3 Historical genealogy of affected families is lengthy and could be substantially shortened given some comments are already integrated in alternate sections.
- The presented manuscript reflects many years of experience and ongoing work. The abstract indicates 13 families in total, with indication of a total of ten identified variants, from the newer families/cases. Figure 1 describes nine families and Table 1 describes 9 families and two individual cases. However, a total of 14 variants are given in Table 2, which may reflect findings for all 14 families studied and the two cases? If this is so, it would be helpful to indicate the four previously identified variants in Table 2.
- The relationship of the mARC1 and MOCOS proteins should be clarified/explained (Figure 2C, and Discussion section text). The formal name MTARC1 should be used or mentioned.
- Section 3.1.2, could be shortened, as some of the interpretation is repeated in the Discussion section.
- Regarding the in vitro variant analyses work for XDH, i) why was a Cys to Ser substitution used to investigate the identified Cys to Phe substitution and ii) why was the thaliana XDH1 gene version used to analyze this variant?
- In the studies to examine the effects of the MOCOS missense variants, i) from which species did the MOCOS-NifS gene fragment originate, that was used for expression in coli, and ii) demonstration of expression and quantitation of the three E. coli-expressed variant polypeptides should be provided and explained. Notably, T349 and P591 are buried residues and may affect protein folding and stability? Also, error bars are not given in the left panel of Figure 3C.
- The assay used in Figure 3D (and lines 519-522) should be explained more clearly. What specifically was tested, as both Moco and molybdopterin binding are indicated (stated as and/or). Are both being measured in a single assay?
- The Discussion section is long, and would benefit from use of sub-sections.
- Given that the authors have noted experience, some discussion on the advantages of early diagnosis would be helpful, including perhaps the prevention of later kidney complications, is this known? Are the late diagnoses due to insufficient investigations, vague early symptoms or perhaps overall variability in presentation between individuals? How do your findings compare with other reports from non-Jewish or Arab populations? Xanthinuria is considered rare, but does your experience suggest/support many missed cases? Further, what surveillance or monitoring is suggested from the presented series with the detection of xanthinuria in pediatric or adult cases, that may extend beyond current standards of care? (These discussions should be qualified as based on experience, but many readers and caregivers may find this useful given the notable size and depth of the patient collection presented.)
- Related to the previous comment, the authors do mention on differences between pediatric and adult patients in the Discussion section, but their interpretations should be clarified, lines 711 – 726. Are the authors suggesting that i) xanthinuria is highly variable, ii) pediatric onset is indication of more severe symptoms and sequelae, or iii) pediatric or early onset is more likely to involve the MOCOS gene (ie. Type II)?
- Could comments be added to highlight what is known about xanthinuria and occurrence of kidney stones or other later stage kidney complications?
- As multiple labs/centers (from different countries) provided data given in Table S2, normal range data and the specific test from the respective lab sources should be provided.
Minor and typographical issues
- Thr349Ileu should read p.Thr349Ile on line 38.
- In the Introduction section, a) the XDH mRNA is indicated to be 80.4kb (line 68), this would appear to be the full gene/transcript size, as RefSeq NM_000379.4 indicates the mRNA is 5715 nucleotides. Similarly, the MOCOS mRNA is indicated to be 84.7kb (line 77), but RefSeq NM_017947.4 indicates the mRNA is 6182 nucleotides.
- The abbreviation HRMA for high resolution melting assay appears several times, but did not appear to be explained. This corresponds to an older style method, and should be explained, suggest at the earliest mention, line 323.
- The word therefor on line 480 should read therefore.
- The word predeominantly on line 499 (in legend to Figure 3) should read predominantly.
- Heterozygous on line 682 should read heterozygotes.
- The words, the fact, on line 698 can be deleted.
- Table 1 and 13 on line 755, should read Tables 1 and S1?
- In Table 1, individual II-2 from F3 should read II-1. And, why are there two blank lines between G1 and G2 cases?
- In Table 1 legend, the abbreviation NSCLC is not explained.
- The legend indicates, highlighted in red, but no red color appears in Table 3.
- What is GRCh38.p13 primary assembly 109 given in the legends to Tables 3 and 4; GRCh38.p13 is from primary assembly 38?
- In Table S1, the second column indicating Gender, should read Sex.
- In Table S1, the indicated sex and age of death of individuals II-1 (F3) and III-1 (F8) are not consistent with data given in Table 1.
Reviewer 2 Report
The authors have painstakingly gathered clinical and molecular information about several large families representative of hereditary xanthinuria. They provide a good clinical picture of the affected individuals, and perform functional studies on the variants which generate important and novel information about the pathogenesis of xanthinuria. Overall, the study is very interesting and potentially quite useful, for clinicians and researchers alike. Here I offer some minor suggestions that, in my opinion, may make the manuscript even more attractive and understandable to the reader: Figure 1/Table 1: each family has been correctly assigned xanthinuria type I or II in the table, but I think it may be helpful to provide a way for the reader to know "at a glance" the family#-XDH or family#-MOCOS associations Section 3.1.2: Either here or, even better, in Table S2, reference values for UA levels (and possibly X and HX levels) should be provided lines 346-348: is this family/patient one of those described in references 26-27? If so, it should be specified in the text. Otherwise, it should be indicated whether these individuals underwent genetic testing for PEPD variants or deletions. lines 358-360: this cannot be inferred from the data presented in table S3. It would be interesting to provide supplementary data supporting the (very logical) assumption that F4 V-5 is homozygous. Furthermore, Table S3 would be more informative if it could provide some data on the similarities between affected individuals from F4 and affected individuals from the left branch (V-1 proband) of F6, in addition to the differences correctly observed with the right branch (IV-17). Section 3.3: This part about historical genealogy of the affected families is fascinating, but it feels like it breaks the flow of the manuscript. Although interesting, I think it is dispensable for the clinical topics discussed in the rest of the manuscript. I would suggest moving this entire section to the supplementary materials, while leaving the historical comments already integrated in the discussion. It would be easier and much more enticing for the reader to move directly from section 3.2, describing the clinical-related results, to section 4, discussing the very same topics. lines 763-767: these considerations feel quite important, and I think they may deserve to be expanded or highlighted a little. For example, it might be worth adding the percentage of asymptomatic homozygous individuals (at a certain age)
Regarding language and form, the manuscript is well written and only requires a check for clerical errors and minor language issues - punctuation, a few sentences that could be improved...
Some specific suggestions: line 36: change "are" with "were" and remove "as follows" line 61: "Management" would be more appropriate than "Treatment" lines 91-92: assuming the 1997-2013 referrals concern both the families and the sporadic cases, it would be more consistent to write "G1 and G2" between parentheses; and it should be "between the years 1997 and 2013, and..." line 115: therefore (last e was missing) line 145: remove "e.g." (it is a predefined cutoff, indeed) line 235, 312, 341, and others: here MOCOS and XDH refer to the genes and should be written in Italics (this is obviously a clerical error, since in most of the text the gene-protein distinction has been made correctly) line 280, 322, and others: HGVS nomenclature for protein variants requires the notation "p.(Aaa###Bbb)" for protein sequences that have not been determined experimentally. Therefore, parentheses should be added where necessary. The notation "(p.(Aaa###Bbb))" is acceptable line 321: the sentence is broken up by parentheses in a strange way. Perhaps it should read as "XDH gene (Table 4 and [24], patient G)."? lines 325-336: The way this section is written makes it difficult to understand (in particular lines 331-336). Besides, these considerations have also been reported in a much better form in the Discussion (lines 696-704), where I think they really belong. I would therefore suggest modifying section 3.1.2 by leaving here only the results and removing the parts regarding their hypothetical reasons (they are more appropriate in the Discussion) line 356: remove comma after "father" line 357: the correct HGVS notation would be NM_000379.3:c.913del (p.(Leu305*)). The letter "C" from "delC" should also be removed from line 341 and elsewhere (abstract, Table 2, etc.) lines 357-358: The mother is "heterozygous". In line with the style used elsewhere, perhaps the past simple rather than the present tense should be preferred: "V-6 and his father were...", "his mother was..." line 363: it should be "The index case, a 4 years old boy, presented with..." line 366: "highly consanguineous" gives an impression of frequency; here "The parents had [or showed] a high degree of consanguinity" would be more appropriate line 385: for consistency, the notation "(p.(Ser624*)) should be used (3-letter aminoacid code and * instead of X for a stop codon have been used elsewhere). Please note that either all nonsense variants should use * instead of X, or all frameshift variants should use X instead of * (this affects multiple instances throughout the manuscript, including Table 2) "... change and a SNV c.1936A>G...": technically they are both SNV. Please specify that c.1936A>G is a common variant (possibly with a due reference to gnomAD or other databases). line 386: it is unclear whether "this variants" refers to c.1871C>G or c.1936A>G line 388: elsewhere in the manuscript, whenever a variant is introduced in the main text its predicted protein outcome is also reported; why is c.2164A>T an exception? line 389: "was a carrier" in both instances lines 401-402: punctuation: "...a third case (V-1), an 8 years old... branch of the family, was..." line 411: SNV: see comment for line 385 line 415: see comment for line 357 line 422: either "the urine was negative for bacterial infections" or "urine culture was negative for bacterial infections" line 427: for consistency, protein notation should be (p.(Trp478*)) lines 429-430: according to HGVS, it should be c.1088_1089del and c.641del. Bases after "del" should also be removed in the other instances throughout the text (lines 437, 443, etc.) line 431: "G1 and G2 were". In general, the same tense - either past or present - should be used to describe patients throughout the manuscript, as appropriate and with due exceptions. lines 437-478: "...in the identification of compound heterozigosity for variants c.1088_1089del (p.(Leu363Profs*16) and c.1771C>T (p.(Pro591Ser)) (Figure S6)." Please note the correct notation for the first variant and correct it throughout the text, e.g. in Table 2 (and check for spaces between c. position and "del") line 443: correct notation would be "c.641del (p.(Pro214Glnfs*4))" line 447: "among the 10"; also, either use "ten" here and "seven" later, or "10" here and "7" later; several style recommendations indicate that consistency overrules the convention of using numbers above 10 and words below 10 lines 448-449: actually most (perhaps all) variants are predicted to undergo nonsense-mediated mRNA decay (NMD). Rather than the lack of essential domains, NMD is the most likely reason for a pathogenic consequence. This should be noted in the manuscript, at least in passing. lines 454-458: I would suggest adding a further tool to corroborate the in silico evaluation, e.g. PolyPhen2, and/or calculating the CADD score - which already takes into account several prediction tools. It would add further value to a well-performed characterization (I did appreciate the further evaluation of residue conservation and accessibility!) line 480: therefore (last e was missing) line 545: lifestyle (single word) line 549: "to a large kindred" (insert "a") line 550: "ruler of Galilee" (remove "the"). Although this region is often styled as "the" Galilee, I believe in the English language the correct form would be just "Galilee" - but please check other sources. line 559: "to an extended Christian-Arab kindred" (insert "an") line 569: "the region of Galilee" (remove "the" - but see comment 550) or, better, "the Galilee region" (in the English language, line 576: "to a very large kindred" (insert "a") line 601: the XO abbreviation should be explained (it is instead explained later, at line 624) line 618: c.449G>T: for clarity, I would suggest using the protein notation here lines 632-633: "has a dramatic effect..." (insert "a"); "which is essential for..." instead of "which is essentially required for..." line 635: "which" instead of "whose" line 639: "contact with the PLP cofactor" (instead of "to") line 648: "has shown" (remove "been") line 652: remove the comma after the parenthesis line 659: I would suggest using a cautious "similar" rather than "identical" line 661: "Besides the..." (remove "to") line 683: remove comma after "families" line 685: remove comma after "F3" and add it after "F8" line 691: add a comma after "related tribes" line 694: add a comma after "families" line 696: remove "run of homozygosity" and leave only "ROH" (it has already been explained in the results section, line 327) lines 703-704: by "this variant prevails in the extended family for many generations", do the authors mean "this variant has been established many generations ago", or "this variant has been prevalent in the extended family for many generations", or else...? line 715: I think the authors intended that "Table 1" and the references should be considered together: "...observation (Table 1 and [12-14])." line 716: using percentages and then absolute numbers to refer to a portion of this percentage is confusing. "8" should be used instead of "42%". Consistent with the form used at line 720, "four" should also be written as "4". line 717: I believe 13 and 14 are references, but the square brackets are missing line 723: "potential" should be used instead of "potentially" line 724: it would be better to repeat "factors" after "environmental" line 727: "pathophysiological" instead of "pathophysiologic" line 728: remove "do" and add "whether" or "as to whether": "the question arises whether the enzymatic deficiencies..." line 730: "On" instead of "At" (On one hand); insert commas after UA and after XDH (or place "the final product of XDH" between spaced dashes: "UA - the final product of XDH - is a reducing...") lines 731 and 737: "humans" (plural) would be preferable instead of "human" (singular) line 740: "(Aox4-/-)" (underscores have mistakenly been inserted instead of minuses) line 741: a noun (e.g. "features") should be inserted after "other" I would also suggest adding a comma after "[7]" and then using "while" instead of "and": "...steatosis [7], while accumulation..." lines 746-753: this sentence is too long. I suggest breaking it up into three sentences, each describing an individual. "She died at the age of 65 years" can then be joined with the previous sentence. Also, lost "to" follow-up (line 748) and "52 years" plural (line 750). line 755: assuming "13" is a reference, the square brackets are missing line 773: in this context, I think "homozygotes" would be more appropriate and precise than "affected" lines 774-775: a parenthesis is missing; maybe the author meant "Since biochemical tests (levels of UA in blood or urine (Peretz et al. unpublished results) or XDH activity in serum [25]) are not useful for detecting carriers,", where "Peretz et al. unpublished results" may also be placed between commas I engaged in this very specific editing in an attempt to improve what I think is already a very good and interesting manuscript. I hope the authors will find it useful. - * - é * - * - lines 331-336: after mentioning "two" possibilities, using "one" and "otherwise" to list them is confusing. Please make this sentence clearer. Furthermore, by "the variant prevails in these large kindred for many generations", do the authors mean "the variant has been established many generations ago"?
Round 2
Reviewer 1 Report
Comments and suggestions for authors:
The revised manuscript has addressed the major concerns. Minor comments, largely typographical in nature, are indicated below.
- Portions of the discussion sections remain redundant with awkward sentences, especially page 17. In this section, it could also be concluded that pathogenicity of missense variants likely involves both polypeptide stability concerns as well as loss of function based on the analyses.
- The conclusions on line 790, page 21, could be strengthened. Suggest … expression studies. … to read … expression studies supporting the pathogenicity of the identified missense changes. … .
- In Figures 3C left panel and 3D, what is indicated with the error bars? How many biological replicates?
- Some of the sections that have been revised include words, phases or sentences with altered font style, example, see lines 599-602, page 17.
- Line 31, page 1; kindred should read kindreds.
- Line 37, page 1; line 289, page 9 and line 474, page 14; Thr349Ileu should read Thr349Ile.
- Line 77, page 2 from Ref Seq NM_017947; 2.7kb should read 6.2 kb.
- Lines 20 and 21, page 5; … origins the gene … should read … origins, the gene … .
- Line 233, page 6; filed should read filled.
- Line 274, page 8; … that his parents … should read … that her parents … .
- Lines 465-466, page 13; inquired should read queried.
- Line 479, page 14; … component, … should read … component (alias mARC1), … .
- Lines 485-486; the sentence, While Thr349 and Pro591 …. is approximately redundant with text on lines 469-470, page 13.
- Line 570, page 16, of should read to.
- Line 691, page 19, enfants should read infants.
- Line 702-703, page 19; suggest that phrase - to occur … and thus are … to read – to be due to underlying genetic cause and thus are …